# Nutrients Intake in Individuals with Hypertension, Dyslipidemia, and Diabetes: An Italian Survey

**DOI:** 10.3390/nu12040923

**Published:** 2020-03-27

**Authors:** Cecilia Guastadisegni, Chiara Donfrancesco, Luigi Palmieri, Sara Grioni, Vittorio Krogh, Diego Vanuzzo, Pasquale Strazzullo, Serena Vannucchi, Graziano Onder, Simona Giampaoli

**Affiliations:** 1Department of Cardiovascular, Endocrine-metabolic Diseases and Aging, Istituto Superiore di Sanita, 00161 Rome, Italy; chiara.donfrancesco@iss.it (C.D.); graziano.onder@iss.it (G.O.); simona.giampaoli@iss.it (S.G.); 2Epidemiology and Prevention Unit, Fondazione IRCCS Istituto Nazionale dei Tumori, 20133 Milan, Italy; 3Associazione Nazionale Medici Cardiologi Ospedalieri (ANMCO), 50121 Florence, Italy; 4Department of Clinical Medicine and Surgery, ESH Excellence Centre of Hypertension, Federico II University of Naples Medical School, 80131 Naples, Italy

**Keywords:** hypertension, diabetes, dyslipidemia, dietary recommendations

## Abstract

The aim of this study is to evaluate whether nutrients intake in an Italian adult population receiving pharmacological treatment for hypertension, dyslipidemia, and diabetes are within the recommended values proposed by dietary guidelines. Cross-sectional data from the Cardiovascular Epidemiology Observatory/Health Examination Survey in 8462 individuals 35–79 years were used. Food consumption was assessed with a self-administered semi-quantitative food frequency questionnaire. Dietary sodium and potassium intakes were measured in 24-hour urine collection. Recommendations from WHO were used for salt and potassium intakes, those from the Diabetes and Nutrition Study Group for diabetes, and those from the European Society of Cardiology for hypertension and dyslipidemia. Salt intake in urine collection of participants receiving treatment for hypertension was 11.1 ± 4.0 g/day for men and 8.6 ± 3.3 g/day for women, higher than recommended. In participants treated for dyslipidemia, mean saturated fat intake was 11.4% and 11.6% total Kcal in men and women respectively, higher than recommended, while cholesterol intake was higher only in men (365.9 ± 149.6 mg/day). In both men and women receiving treatment for diabetes, mean intake of saturated fats (12.3% and 12.2% of total Kcal), simple carbohydrates (17.5% and 19.8% of total Kcal) and cholesterol (411.0 ± 150.4 and 322.7 ± 111.1 mg/day) were above the recommendations, while fiber intake was below (19.5 ± 6.3 and 17.5 ± 6.2 mg/day). Overall, 70% to 80% of participants treated for these conditions received advice from family doctors on dietary management; however, nutrition is far from being optimal.

## 1. Introduction

Hypertension, dyslipidemia, and diabetes are known risk factors of cardiovascular diseases (CVD) and a healthy diet is considered as the first approach to treat these conditions [1]. In particular, reduction of salt intake is considered critical for treatment of hypertension, given its positive effect on blood pressure (BP) in hypertensive individuals [2,3]. Also, potassium intake has an effect on lowering BP in both hypertensive and normotensive individuals [2,3].

Similarly, reduction of saturated fatty acids (SFA) intake is fundamental for the treatment of dyslipidemia [4]. It is recognized that SFA increase cardiovascular risk by raising plasma low-density lipoprotein (LDL) cholesterol whereas polyunsaturated fatty acids (PUFA) and monounsaturated fatty acids (MUFA) lower this parameter [5,6]. Finally, nutrition therapy is an integral component of diabetes management and even modest weight loss improves insulin sensitivity in individuals with type 2 diabetes [7,8]. In addition, high fiber intake and low refined carbohydrates are one of the most important interventions for improving glycemic control in individuals with diabetes. The fiber intake recommended for the diabetic population is indeed double that recommended for the general population [9].

The aim of the present study is to evaluate the dietary habits of an Italian population receiving pharmacological treatment for hypertension, dyslipidemia, and diabetes in comparison with the general population not receiving such treatments. In addition, we have assessed whether macronutrients intake and salt and potassium excretion is within the recommended values proposed by current dietary guidelines and we have evaluated how many participants with these conditions receive guidance on dietary recommendations from their family physician.

## 2. Materials and Methods

### 2.1. Participants

For the present study we used data from the Cardiovascular Epidemiology Observatory/Health Examination Survey, which was conducted in 2008–2012 in the adult Italian general population [10]. The aim of the survey was to describe lifestyle habits (habitual diet, physical activity, alcohol consumption) and the prevalence of high-risk conditions for CVD. Two hundred and twenty men and women per 1.5 million general population aged 35–79 years and living in 20 Italian regions were randomly selected. An information letter about the project was sent to all the selected participants to explain the research purposes and obtain a signed informed consent to participate. Trained personnel administered questionnaire collecting information on the main characteristics of the participants, including demographics, anthropometric measures and drug treatment, and participants filled out self-administered dietary information. The current study population consisted of people taking drugs for hypertension, dyslipidemia, or diabetes. This variable was assessed by study personnel by specific questions (i.e., ‘Are you using any drugs for treatment of …?’). Moreover, study personnel asked the participants, whether their family physician had given advice regarding lifestyle and dietary changes to treat hypertension, dyslipidemia, and diabetes. The Italian National Institute of Health ethical committee approved the survey on 11 November, 2009.

### 2.2. Assessment of Nutrients Intake

Dietary information was collected by the self-administered Italian version of the food frequency questionnaire (FFQ) of the European Prospective Investigation into Cancer (EPIC), which focused on diet as a major determinant of health. Accuracy and validation of dietary questionnaires was of paramount importance as it was to be applied in several countries and to thousands of study participants [11].The FFQ was designed to capture eating behaviors in the Italian population [12]. Food items were linked using specifically designed software [13] to Italian Food Tables [14] to obtain estimates of daily intake of 37 macro- and micronutrients plus energy (not all nutrients are shown in the tables). Carbohydrates, protein, and fat intakes were classified as percentage of total daily energy intake, dietary cholesterol as mg/day, and dietary fiber, sodium, and potassium as g/day. In addition to using the EPIC questionnaire to assess dietary sodium and potassium intakes, levels were also objectively measured through a 24-hour urine collection in a random sample of 200 people in each region. Participants were provided with a SARSTED plastic container of 3 L with thymol added for urine preservation and assayed centrally.

WHO recommendations were used as the reference for normal salt and potassium intakes [3]. Recommendations from the Diabetes and Nutrition Study Group (DNSG) of the European Association for the Study of Diabetes, endorsed by the Italian Diabetes Society (SID), were used as reference for intake of other nutrients in diabetes [8,15], and European Society of Cardiology (ESC) recommendations were used for hypertension and dyslipidemia [1].

### 2.3. Study Sample

Of the 8696 men and women aged 35–79 years considered in this analysis, 464 (5.3%) did not complete the dietary questionnaire making it impossible to calculate the total energy intake. We also excluded 176 (2.0%) persons in whom the ratio of total energy intake to basal metabolic rate was at either extreme of the distribution (i.e., energy intake below 400 kcal/day or higher than 5000 kcal/day), leading to a final sample of 8056 participants. For the purpose of the present study, the sample was categorized according to the use of drugs for treatment of hypertension, dyslipidemia, or diabetes. This was done to assess adherence to guidelines and recommendations in persons already receiving pharmacological treatment for these conditions. Urine excretion of sodium and potassium was available for 3513 participants out of 8056; 24-hour urine volume was equal or over 500 ml and its creatinine content referred to body weight was found to be within mean minus/plus 2 standard deviation from the population mean.

### 2.4. Statistical Analysis

Separate analyses were done for treatment of hypertension (present or not present), hypercholesterolemia (present or not present), and diabetes (present or not present). Each participant may have been included in all three analyses if criteria of inclusion, according to treatment, were met. Hence the number of total persons included in each analysis (hypertension, dyslipidemia, diabetes) was different.

Data are presented, separately for men and women, as mean, standard deviations, and prevalence, with 95% Confidence Intervals (C.I.), standardized by age using the 2013 European standard population age distribution, equal for both men and women. Multivariate regression models were performed for each nutrient; BMI and total energy intake were used to compare treated and not treated groups. Models were adjusted for age, educational level, smoking habit, physical inactivity, and overweight/obesity. Educational level was selected as a proxy for socioeconomic status subdividing participants between low educational level (primary/middle school) and high educational level (high school/university). Smoking was defined as a person smoking one or more cigarettes per day. Physical activity during leisure time was collected through a questionnaire including four levels of exercise (sedentary, mild, moderate, heavy). However, in this analysis, physical activity was considered only as sedentary and non-sedentary. Height and weight were measured following standardized methods; overweight or obesity was defined as BMI above 25 kg/m^2^. Further details about procedure and methods used to collect data can be found in the previous publication [10]. Chi-squared test was performed to compare prevalence of treated and not treated groups (Appendix A). Box plots were used to show the distribution of total energy intake by gender and treatment (Appendix A). Two sided p-values equal or less than 0.05 were considered statistically significant. Statistical analyses were performed using SAS (9.4 version).

## 3. Results

Table 1 presents nutritional intake in men and women treated for hypertension as compared with those not receiving treatment. Nutritional recommendations issued by ESC and/or WHO are reported for comparison.

There were no significant differences between participants treated and untreated for hypertension in the intake of total lipids, cholesterol, and fiber, while in the treated group a significant lower intake in both total and simple carbohydrates in men and only simple carbohydrates in women was observed. Both men and women of the treated group had a higher BMI as compared with the untreated group and energy intake was slightly reduced in treated as compared with not treated men. Salt excretion in participants receiving treatment for hypertension was higher than untreated participants (11.1 ± 4.0 g/day vs. 10.6 ± 4.1 g/day in men and 8.6 ± 3.3 g/day vs. 8.1 ± 3.3 g/day in women) although not significant. Overall, 70% of the participants treated for hypertension reported to have been advised to reduce salt intake by family doctor as compared to only 6% of the untreated population.

The nutritional intakes of the population under treatment for dyslipidemia together with ESC/WHO recommendations are presented in Table 2.

Intake of SFA was significantly lower in the treated as compared with the untreated sample as well as intake of total cholesterol. Both in men and women, the treated group had a higher BMI and a significant lower energy intake as compared with the untreated group. Saturated fat intake was higher than suggested by ESC recommendations in both men and women. Among treated women, mean cholesterol intake was below the threshold suggested by ESC recommendations, while mean value among men was above the threshold. A high percentage of the population treated for hypercholesterolemia (76.4% men 76.6% women) received recommendations for reducing plasma lipids whereas only 22.1% of men and 21.7% of women untreated received this advice.

Nutrients intake according to diabetes treatment, along with the recent nutritional recommendations issued by DNSG, are presented in Table 3.

In both men and women, treated participants had a higher intake of PUFA and a lower intake of total carbohydrates and simple carbohydrates. Gender differences were observed in the intake of SFA and cholesterol, where treatment was associated with a higher intake of these nutrients in men and with a lower intake in women. Both men and women treated for diabetes had a significantly higher BMI than the untreated population. Total energy intake was significantly lower in the treated than in the untreated sample. In both treated men and women, mean intake of lipids, SFA, simple carbohydrates, and cholesterol was above the threshold suggested by DNSG guidelines, while fiber intake was below the suggested threshold. A high percentage of persons treated for diabetes (80% of men and 86% of women) had received advice from family doctors on lifestyle and nutritional changes.

## 4. Discussion

These data from a national Italian survey show that dietary intake in persons treated for hypertension, dyslipidemia, and diabetes is far from being optimal. Participants treated for hypertension have a higher salt consumption than untreated participants, the mean value of SFA intake in participants treated for dyslipidemia is higher than the <10% recommended level, and only women have the recommended <300 mg/day cholesterol intake. Participants treated for diabetes have mean fiber consumption much lower than recommended.

Data on the dietary salt intake in this Italian population are in line with the results reported in the survey of the MINISAL-GIRCSI study, which showed that salt intake in Italy is high [16]. We showed that excretion of salt by the population under treatment for hypertension was higher than the general population. This finding is in contrast with data from the MINISAL-SIIA study of Italian hypertensive patients that showed a slightly lower sodium intake in comparison with the general population [17]. Indeed, our finding might be explained by the presence of a direct association between BMI and average salt intake, as the hypertensive treated population had a higher BMI than the general population [18]. It was hypothesized that high BMI might be associated with high salt intake due to a greater food and calories consumption. As calories intake is slightly reduced in our hypertensive population, the diuretic treatment for hypertension might have increased sodium excretion [19].

Confirming the results of the previous surveys, potassium intake was, on average, much below the dietary recommended intake of 3.9 g/day. A diet rich in potassium has an antihypertensive effect through various possible mechanisms and, therefore, professional organizations have made recommendations to increase the average intake of potassium as, in many countries, it is relatively low [20]. A concomitant increase in potassium, magnesium, calcium, and dietary fiber and a decrease in salt intake is currently the most recommended essential lifestyle intervention for controlling BP; these recommendations are outlined in the Dietary Approaches to Stop Hypertension [21].

Saturated lipids intake in participants treated for dyslipidemia is higher than recommended in the guidelines. Dietary SFA have a potent plasma LDL cholesterol raising effect, which is a causal risk factor for CVD. The substitution of dietary SFA with PUFA consistently reduces CVD events [4]. Several observational studies and meta-analyses have not demonstrated a significant correlation between SFA consumption and CVD events, but the isoenergetic replacement with PUFA has shown reduced CVD risk [5,6]. Thus, control of LDL cholesterol through reduction of SFA intake remains a cornerstone for management of dyslipidemia and prevention of cardiovascular events [22].

Our data confirm previous findings suggesting that compliance with nutritional recommendations for diabetes is extremely low and that carbohydrates intake in persons receiving treatment for diabetes is lower than the general population [23,24]. In persons with diabetes, complex carbohydrates intake is beneficial and fat-rich and simple sugar-containing foods should be avoided [25].

Despite the fact that between 70% and 80% of the population treated for hypertension, dyslipidemia, and diabetes received suggestions by family doctors on how to change dietary habits, nutrients intake was still not optimum on average. Previous studies have shown that specific intensive interventions based on detailed individualized counseling for lifestyle modification are necessary to achieve dietary modifications [26,27]. This kind of intervention is more effective than metformin in preventing diabetes incidence [28]. It has been shown that individual guidance for physical activity and sessions with nutritionists in persons with impaired glucose tolerance leads to substantial weight loss and a reduction in diabetes incidence [29]. In addition to the individualized interventions, a community-based intervention can also achieve changes in nutrition through the involvement of the local food industry for reducing and modifying the fat and salt content of commonly eaten foods. In Finland, a nation-wide population-based project aimed at improving nutrition through education and social media was shown to impact on dietary habits and to reduce serum cholesterol and BP levels [30].

The major strength of our study is that it is based on a population-based survey, the Cardiovascular Epidemiology Observatory/Health Examination Survey that covers all Italian regions. A major limitation is represented by the limited availability of 24-hour urine samples and the need to rely on the EPIC questionnaire to assess sodium and potassium in the whole study sample. The EPIC questionnaire is considered to be of high standard of accuracy and country specificity. However, the salt commonly added while cooking and the discretionary salt added at table is not calculated in the EPIC questionnaire [12].

## 5. Conclusions

In conclusion, dietary changes should be considered as the first approach for treatment of hypertension, diabetes, and dyslipidemia, in order to reduce the risk of CVD, but nutritional intakes in persons with these conditions is far from being optimal. Family doctors often advise persons with these conditions on correct dietary approach, but these recommendations are poorly followed.

## Figures and Tables

**Table 1 nutrients-12-00923-t001:** Nutrients composition of diet in men and women receiving and not receiving treatment for hypertension. Men and women aged 35–79 years from the Osservatorio Epidemiologico Cardiovascolare/Health Examination Survey of the CUORE Project 2008–2012.

Nutrients	Men					Women					ESC/WHO Recommendations
	Not Treated for Hypertension *n* = 2749		Treated for Hypertension *n* = 1280			Not Treated for Hypertension *n* = 2837		Treated for Hypertension *n* = 1174			
	Mean ± SD	Mean 95% C.I.	Mean ± SD	Mean 95% C.I.	Sign.	Mean ± SD	Mean 95% C.I.	Mean ± SD	Mean 95% C.I.	Sign.	
**Proteins (% total Kcal)**	16.0 ± 2.5	15.9–16.1	16.3 ± 2.6	16.1–16.4	*	16.5 ± 2.6	16.4–16.6	17.0 ± 2,7	16.8–17.1	ns	10–20%
**Total Lipids (% total Kcal)**	33.8 ± 5.7	33.7–34.1	34.3 ± 6.4	33.9–34.6	ns	36.5 ± 5.7	36.3–36.7	36.5 ± 6.2	36.1–36.9	ns	20–35%
**Saturated fat (% total Kcal)**	11.8 ± 2.5	11.7–11.9	11.8 ± 2.6	11.7–11.9	ns	12.7 ± 2.5	12.6–12.8	12.4 ± 2.6	12.2–12.5	*	<10%
**Polyunsaturated fat (% total Kcal)**	3.9 ± 0.9	3.8–3.9	4.0 ± 1	3.9–4	ns	4.2 ± 0.9	4.1–4.2	4.3 ± 1.1	4.2–4.3	ns	<10%
**Carbohydrate (% total Kcal)**	46.9 ± 7.9	46.6–47.2	46.0 ± 8.6	45.5–46.5	**	47.6 ± 7.7	47.3–47.9	47.6 ± 8.1	47.1–48	ns	45–60%
**Simple carbohydrates (% total Kcal)**	19.6 ± 7.0	19.4–19.9	18.6 ± 7.0	18.2–19.0	**	22.6 ± 7.2	22.3–22.8	21.4 ± 7.1	21–21.8	**	<10%
**Cholesterol (mg/day)**	384.8 ± 164.5	378.7–391	380.5 ± 144.6	372.6–388.4	**	336.7 ± 139.9	331.6–341.9	330.4 ± 135.6	322.6–338.1	ns	<300
**Fiber (g/day)**	19.4 ± 7.4	19.2–19.7	19.9 ± 6.8	19.5–20.2	*	18.5 ± 7.1	18.2–18.7	18.0 ± 6.8	17.6–18.4	ns	30–45
**Sodium (g/day)**	2.3 ± 1.1	2.3–2.4	2.5 ± 1.0	2.4–2.5	ns	1.9 ± 0.9	1.9–1.9	1.9 ± 0.9	1.8–1.9	ns	<2
**Potassium (g/day)**	3.3 ± 1.1	3.2–3.3	3.3 ± 1.0	3.2–3.3	**	3.0 ± 1.0	3–3.1	2.9 ± 1.0	2.9–3	ns	>4
**Salt g/day (24hr urine)**	10.6 ± 4.1	10.4–10.8	11.1 ± 4.0	10.8–11.4	ns	8.1 ± 3.3	7.9–8.3	8.6 ± 3.3	8.3–8.8	ns	<5
**Potassium g/day (24hr urine)**	2.5 ± 0.7	2.4–2.5	2.5 ± 0.8	2.4–2.5	ns	2.2 ± 0.7	2.1–2.2	2.2 ± 0.7	2.1–2.2	ns	>3.9
**Body Mass Index (kg/m^2^)**	27.1 ± 4.1	27–27.3	29.4 ± 4.1	29.2–29.7	***	26.3 ± 4.9	26.1–26.5	30.5 ± 5.9	30.1–30.8	***	<25
**Energy (Kcal/day)**	2297 ± 770	2268–2326	2276 ± 697	2238–2314	**	1899 ± 647	1875–1923	1816 ± 628	1781–1852	ns	

Values are means ± standard deviation. C.I. Confidence Intervals. ESC (European Society of Cardiology). Sign = statistical significance of treatment coefficient in a multivariate regression model adjusted by age, educational level, smoking habit, physical inactivity, and overweight/obesity. *** *p* < 0.0001; ** *p* < 0.01; * *p* < 0.05; ns not statistically significant. Data are standardized by gender and age using the 2013 European population. Number of persons with available salt and potassium g/day (24 hour urine): 1219 untreated and 564 treated men and 1214 untreated and 510 treated women.

**Table 2 nutrients-12-00923-t002:** Nutrients composition of the diet in men and women receiving and not receiving treatment for dyslipidemia. Men and women aged 35–79 years from the Osservatorio Epidemiologico Cardiovascolare/Health Examination Survey of the CUORE Project 2008–2012.

Nutrients	Men					Women					ESC/WHO Recommendations
	Not Treated for Dyslipidemia *n* = 3455		Treated for Dyslipidemia *n* = 540			Not Treated for Dyslipidemia *n* = 3456		Treated for Dyslipidemia *n* = 537			
	Mean ± SD	Mean 95% C.I.	Mean ± SD	Mean 95% C.I.	Sign.	Mean ± SD	Mean 95% C.I.	Mean ± SD	Mean 95% C.I.	Sign.	
**Proteins (% total Kcal)**	16.0 ± 2.4	15.9–16.1	16.4 ± 2.9	16.2–16.6	**	16.6 ± 2.7	16.5–16.7	16.7 ± 2.8	16.5–16.9	ns	10–20%
**Total Lipids (% total Kcal)**	33.9 ± 5.9	33.7–34.1	33.2 ± 6.0	32.7–33.7	ns	36.6 ± 5.8	36.4–36.8	36.0 ± 6.0	35.5–36.5	*	20–35%
**Saturated fat (% total Kcal)**	11.8 ± 2.5	11.8–12	11.4 ± 2.6	11.2–11.7	**	12.7 ± 2.5	12.6–12.8	11.8 ± 2.6	11.5–12	***	<10%
**Polyunsaturated fat (% total Kcal)**	3.9 ± 0.9	3.9–3.9	3.9 ± 0.9	3.8–3.9	ns	4.2 ± 1.0	4.1–4.2	4.2 ± 1.1	4.1–4.3	ns	<10%
**Carbohydrate (% total Kcal)**	46.8 ± 8.1	46.5–47	46.6 ± 8.3	45.9–47.3	ns	47.5 ± 7.8	47.3–47.8	47.3 ± 8.2	46.6–48	ns	45–60%
**Simple carbohydrates (% total Kcal)**	19.5 ± 7.0	19.2–19.7	18.8 ± 6.7	18.2–19.4	*	22.3 ± 7.2	22–22.5	20.9 ± 7.0	20.3–21.5	ns	<10%
**Cholesterol (mg/day)**	382.9 ± 159.3	377.6–388.3	365.9 ± 149.6	353.3–378.5	**	340.1 ± 139.8	335.4–344.7	294.9 ± 127.9	284–305.7	***	<300
**Fiber (g/day)**	19.4 ± 7.3	19.1–19.6	19.0 ± 7.0	18.4–19.6	ns	18.5 ± 7.1	18.2–18.7	17.8 ± 6.5	17.3–18.4	ns	30–45
**Sodium (g/day)**	2.4 ± 1.1	2.4–2.4	2.3 ± 1.0	2.2–2.4	**	1.9 ± 0.9	1.9–2.0	1.7 ± 0.8	1.7–1.8	**	<2
**Potassium (g/day)**	3.3 ± 1.0	3.2–3.3	3.2 ± 1.1	3.2–3.3	ns	3.0 ± 1.0	3.0–3.0	2.9 ± 1.0	2.8–3	ns	>4
**Salt g/day (24hr urine)**	10.7 ± 4.1	10.4–10.9	11.3 ± 4.1	10.8–11.8	ns	8.3 ± 3.3	8.1–8.4	7.8 ± 3.0	7.4–8.2	ns	<5
**Potassium g/day (24hr urine)**	2.5 ± 0.7	2.4–2.5	2.5 ± 0.8	2.4–2.6	ns	2.2 ± 0.7	2.1–2.2	2.0 ± 0.7	2–2.1	ns	>3.9
**Body Mass Index (kg/m** **^2^** **)**	27.6 ± 4.1	27.5–27.7	28.5 ± 4.1	28.1–-28.8	**	27.1 ± 5.4	27–27.3	28.0 ± 5.2	27.5–28.4	ns	<25
**Energy (Kcal/day)**	2283 ± 649	2262–2305	2185 ± 726	2124–2246	**	1899 ± 754	1874–1924	1755 ± 585	1706–1805	**	

Values are means ± standard deviations. C.I. = Confidence Intervals. ESC (European Society of Cardiology). Sign = statistical significance of treatment coefficient in a multivariate regression model adjusted by age, educational level, smoking habit, physical inactivity, and overweight/obesity. *** *p* < 0.0001; ** *p* < 0.01; * *p* < 0.05; ns not statistically significant. Data are standardized by gender and age using the 2013 European population. Number of persons with available salt and potassium g/day (24 hour urine): 1530 not treated and 235 treated men, 1474 not treated and 241 treated women.

**Table 3 nutrients-12-00923-t003:** Nutrients composition of the diet in men and women receiving and not receiving treatment for diabetes. Men and women aged 35–79 years from the Osservatorio Epidemiologico Cardiovascolare/Health Examination Survey of the CUORE Project 2008–2012.

Nutrients	Men					Women					DNSG/WHO Recommendations
	Not Treated for Diabetes *n* = 3725		Treated for Diabetes *n* = 308			Not Treated for Diabetes *n* = 3820		Treated for Diabetes *n* = 194			
	Mean ± SD	Mean 95% C.I.	Mean ± SD	Mean 95% C.I.	Sign.	Mean ± SD	Mean 95% C.I.	Mean ± SD	Mean 95% C.I.	Sign.	
**Proteins (% total Kcal)**	16.0 ± 2,5	15.9–16.0	17.5 ± 2.8	17.2–17.8	***	16.6 ± 2.6	16.5–16.6	17.8 ± 3.0	17.3–18.2	***	10–20%
**Total Lipids (% total Kcal)**	33.8 ± 5.9	33.6–34.0	35.8 ± 6.2	35.1–36.5	***	36.5 ± 5.8	36.3–36.7	37.5 ± 7.0	36.5–38.5	*	25–35%
**Saturated fat (% total Kcal)**	11.8 ± 2.6	11.7–11.9	12.3 ± 2.6	12–12.6	*	12.7 ± 2.6	12.6–12.7	12.2 ± 2.5	11.8–12.5	ns	<10%
**Polyunsaturated fat (% total Kcal)**	3.9 ± 0.9	3.8–3.9	4.2 ± 0.9	4.1–4.3	***	4.2 ± 1.0	4.1–4.2	4.5 ± 1.0	4.3–4.6	*	<10%
**Carbohydrate (% total Kcal)**	46.8 ± 8.1	46.6–47.1	44.2 ± 8.1	43.3–45.1	**	47.6 ± 7.8	47.4–47.9	45.5 ± 9.5	44.1–46.8	**	45–60%
**Simple carbohydrates (% total Kcal)**	19.5 ± 7.0	19.3–19.8	17.5 ± 6.2	16.8–18.2	***	22.4 ± 7.1	22.2–22.6	19.8 ± 7.2	18.8–20.8	***	<10
**Cholesterol (mg/day)**	380.1 ± 159.1	375–385.2	411.0 ± 150.4	394.2–427.7	ns	338.1 ± 140.0	333.6–342.5	322.7 ± 111.1	307–338.3	**	<200
**Fiber (g/day)**	19.3 ± 7.3	19.1–19.6	19.5 ± 6.3	18.8–20.2	ns	18.5 ± 7.1	18.2–18.7	17.5 ± 6.2	16.7–18.4	ns	>40
**Sodium (g/day)**	2.4 ± 1.1	2.3–2.4	2.5 ± 1.1	2.4–2.7	ns	1.9 ± 0.9	1.9–1.9	1.8 ± 0.8	1.7–1.9	*	<2
**Potassium (g/day)**	3.3 ± 1.1	3.2–3.3	3.3 ± 1.0	3.2–3.4	ns	3.0 ± 1.0	3.0–3.0	3.0 ± 1.0	2.8–3.1	ns	>4
**Salt g/day (24h urine)**	10.6 ± 4.1	10.4–10.8	10.9 ± 3.6	10.3–11.5	ns	8.2 ± 3.3	8.1–8.4	9.2 ± 2.8	8.6–9.8	ns	<5
**Potassium g/day (24h urine)**	2.5 ± 0.8	2.4–2.5	2.7 ± 0.8	2.6–2.8	ns	2.2 ± 0.7	2.2–2.2	2.3 ± 0.7	2.2–2.5	ns	>3.9
**Body Mass Index (kg/m** **^2^** **)**	27.6 ± 4.1	27.4–27.7	29.8 ± 4.3	29.3–30.2	***	27.0 ± 5.2	26.8–27.2	32.1 ± 5.7	31.3–32.9	***	<25
**Energy (Kcal/day)**	2282 ± 754	2258–2306.4	2205 ± 664	2131–2279.3	***	1896 ± 647	1875.5–1916.5	1735 ± 518	1662.5–1808.3	***	

Values are means ± standard deviations. C.I. = Confidence Intervals. DNSG (Diabetes and Nutrition Study Group). Sign = statistical significance of treatment coefficient in a multivariate regression model adjusted by age, educational level, smoking habit, physical inactivity, and overweight/obesity. *** *p* < 0.0001; ** *p* < 0.01; * *p* < 0.05; ns not statistically significant. Data are standardized by gender and age using the 2013 European population Number of persons with available salt and potassium g/day (24 hour urine): 1657 not treated and 128treated men, 1639 not treated and 85 treated women.

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
