# Peer review of "Nutrients Intake in Individuals with Hypertension, Dyslipidemia, and Diabetes: An Italian Survey"

_nutrients, 2020, doi:10.3390/nu12040923_

Round 1

Reviewer 1 Report

The authors evaluate the dietary intake patterns across several Italian regions by utilizing a food frequency questionnaire in order to assess the adherence to the recommended dietary guidelines among individuals on treatment for diabetes, hypertension, and dyslipidemia. The authors find that the quality of the diet consumed by individuals under treatment is sub-optimal and continues to be so despite receiving dietary advice from their doctors.

This study is definitely of relevance to the present-day world as diet plays a crucial role in determining the overall health of the general population and chronic, non-communicable diseases such as diabetes, dyslipidemia, and hypertension continue to be highly prevalent.

There are sections of the manuscript that need improvement and the suggestions are mentioned below:

The final paragraph of the “Introduction” section should be written in the past tense. Similarly, the ‘Statistical methods’ should also be in the past tense.

Under “Methods”, the authors mention that the food frequency questionnaire obtained data on 37 different nutrients but many of them are not represented in the tables. I suggest the authors to either delete this sentence or mention, maybe in brackets, mention that not all nutrient intake is shown in the tables.

In the tables, for several values, commas have been used to indicate decimals instead of full stop/period. Please correct this in all the tables and utilize decimal points instead of commas to denote decimal values.

Please check the title of Table 3 which says “Men and women aged 35-79 years from Table 2008”.

Also, the 24-hour urine value for sodium is mentioned as 'sodium chloride' in the tables. This needs to be changed to 'sodium'.

In the second paragraph of the discussion, the authors mention that the elevated 24-hour urine sodium excretion was higher in the treated group. It must be noted that his increased sodium excretion in the treatment groups could possibly be a result of diuretic (especially thiazides) therapy, which is often the first-line treatment for hypertension. If the authors have data on diuretic therapy in their study population, they could perform a multiple regression analysis to check for the effects of diuretics on 24-hour urine sodium excretion. If data is not available, then the authors should at least mention the potential effect of diuretics on sodium excretion in the discussion. In addition, the authors mention that they assessed daily salt intake by using 24-hour sodium excretion as a surrogate. However, based on Table 1, women seem to in fact have mean sodium consumption (g/day) of 1.9g, which is within the WHO guidelines of <2 g/day. This has not been discussed by the authors in the article.

In the third paragraph of discussion, in the sentence, “A concomitant increase in potassium, magnesium, calcium, and dietary fiber and a decrease in intake”, I think the authors mean to mention “decrease in salt intake”. The word "salt" is missing.

In the fifth paragraph of discussion, the statement, “In persons with diabetes a diet rich in carbohydrates is more beneficial than a diet rich in fats” is highly misleading. In patients with diabetes, cutting down simple carbohydrates is crucial. The authors should modify this sentence or delete the sentence. The modification can be made on the lines of something like this: “In persons with diabetes, complex carbohydrate intake is beneficial and fat-rich and simple sugar-containing foods should be avoided.”

It is also important that the authors briefly describe the strengths and limitations of the EPIC food frequency questionnaire in the manuscript (does it accurately predict daily caloric or macronutrient intake? does it not gather information on certain food items that might be rich in salt? and so on).

Reviewer 2 Report

This study was conducted using the data from the Cardiovascular Epidemiology Obesity/Health Examination Survey in Italy, which seems to provide a high quality data set. The study focused on the “worn-out” hypothesis, the comparison of nutrients intake between patient and non-patient groups. In addition, it is required to adjust for age and potential confounders to compare two groups for scientific evidence. The analysis should be further refined by using basic techniques frequently used in nutrition epidemiology. Further details are following.

  1. The raw comparison of major nutrients intake between patients and non-patient groups is rather a blunt theme, may not be for an original article in scientific journals.
  2. A background characteristics of study participants should be presented according to treatment status. Namely, summary statistics of age, BMI, socioeconomic status such as education, physical activity, employment status, and cigarette smoking status and so on, is needed. Without knowing the distribution difference of each characteristics, there is no way to know whether they affected the study results.
  3. The statistical test to compare the two groups should be adjusted for potential confounders. Performing t-test is not sufficient to control for confounders.
  4. The distribution of total calorie intake should be presented.
  5. The analysis of each nutrients intake should be adjusted for total calorie intake.
  6. The intake of major vitamins and minerals should be analyzed.
  7. How each treatment group was defined should be described more precisely. The description in line 59-72 is too brief, and hence the validity of definition cannot be judged.
  8. Please describe whether the study protocol was approved by the ethics committee or IRB.

Round 2

Reviewer 2 Report

Authors have made reasonable corrections. Still, I have an impression that the modifications are superficial. The summary statistics is equally as important as the comparison test and provides valuable information to the readers, who are presumably family physicians.  There is abudant evidence supporting that vitamins and minerals are the protective/risk factors for major chronic diseases. The important materials should be presented and discussed in the main article, not in the supplement materials. Besides, box plot is not the best way to present data with this large number of subjects.